# Kinematic Comparison of Different Types of Start Technique in Bi-Finswimming

**DOI:** 10.3390/jfmk10040384

**Published:** 2025-10-02

**Authors:** Gregory Kalaitzoglidis, Konstantinos Papadimitriou, Ioannis Kostoulas, Anastasios Papadopoulos, George Tsalis

**Affiliations:** 1School of Physical Education and Sport Science at Serres, Aristotle University of Thessaloniki, 54124 Thessaloniki, Greece; gkalaitzoglidis@gmail.com; 2Department of Nutritional Sciences and Dietetics, School of Health Sciences, International Hellenic University, 57400 Thessaloniki, Greece; kpapadimitriou@ihu.gr; 3Department of Physical & Cultural Education, Hellenic Army Academy, 16673 Athens, Greece; jkost@otenet.gr; 4School of Physical Education and Sport Science, Democritus University of Thrace, 69100 Komotini, Greece; swimtechniq@gmail.com

**Keywords:** finswimming, OSB11, diving, dominant leg, preferred leg

## Abstract

**Objective:** This study aimed to experimentally investigate the biomechanical and performance differences between the grab start (GS) and the kick start (KS) with each leg on the kickplate (KSR, KSL) in Bi-Finswimming (BFS). It focused on the effect of foot placement on the starting block, equipped with an adjustable, inclined rear kickplate (Omega, OSB11), to determine potential performance advantages and contribute evidence-based recommendations for optimizing start techniques in competitive BFS. **Methods:** Thirteen national-level finswimmers (seven males, six females; age: 17.7 ± 2.1 years) voluntarily participated. Each athlete performed two trials of three start techniques (GS, KSR, KSL) over three days in a randomized order. Four synchronized cameras recorded video data. Performance metrics (time to 5 m (T5), 15 m (T15), 25 m (T25), reaction time, block time (BT), flight time (FT), and entry characteristics) along with joint angles (hip, knee, ankle), were analyzed using Kinovea software (v. 2024.1). A two-way repeated measures ANOVA (start type × gender) was conducted to analyze performance metrics, and a paired-sample t-test assessed differences in joint angles. Also, correlations between dependent (type of start) and independent variables (start-examined variables) were examined through bivariate Pearson’s r analysis. **Results:** No significant gender differences were found (*p* > 0.05). Significant differences emerged between the starting techniques, with KS showing faster T5, T15, and T25 (*p* < 0.001, η^2^_p_ = 0.6; *p* < 0.001, η^2^_p_ = 0.5; *p* < 0.05, η^2^_p_ = 0.3, respectively). BT was significantly longer in GS compared to KS (*p* < 0.001, η^2^_p_ = 0.8), while FT was shorter in GS (*p* = 0.002, η^2^_p_ = 0.4). Faster T5, T15, and T25 were associated with increased flight distance and longer FT in KSL. **Conclusions:** The kick start generally outperforms the grab start, especially in block time, in Bi-Finswimming. These preliminary results suggest that it could be considered for future discussion regarding potential legalization by the World Underwater Federation, pending further research.

## 1. Introduction

Finswimming (FS) is a competitive aquatic sport that requires specialized technical skills and equipment. A notable discipline within FS is Bi-Finswimming (BFS), which was officially incorporated into the competition program of the World Underwater Federation in 2007 [1]. BFS primarily employs a freestyle swimming technique, with athletes utilizing a pair of standardized fins composed of homogeneous rubber and a snorkel [1]. The start phase is pivotal among the critical determinants of performance in competitive conventional swimming (CS) and FS, particularly in sprint events, where minor time differences can significantly influence race outcomes [2]. Since the introduction of the Omega OSB11 starting platform, equipped with an inclined kickplate at the rear, considerable research in competitive swimming (CS) has focused on analyzing and comparing starting techniques. Particular attention has been given to the differences among the traditional grab start (GS), the track start (TS), which involves a split stance on the block with one foot forward gripping the front edge and the other positioned backward for leverage, and the kick start (KS), where the rear foot is placed on the inclined kickplate [3,4]. Nevertheless, the initiation on the OSB11 starting platform is occasionally referred to as either a track start (TS) or a kick start (KS) in the relevant literature [5,6].

Although BFS performance is influenced by unique factors such as snorkel use, fins, and underwater dynamics [1], the present study does not aim to address the entirety of these elements. Instead, our focus is restricted to the start phase, as the effectiveness of the starting technique plays a decisive role in competitive outcomes. Previous research in CS has demonstrated that differences in start technique can significantly impact performance [5,6]; however, it remains unclear whether similar patterns apply in BFS. By isolating the start phase and comparing two distinct starting techniques, our study provides targeted insights that may contribute to a better understanding of performance optimization in BFS without confounding influences from other race components.

Previous research in CS has thoroughly examined start performance, highlighting crucial variables such as reaction time (RT), block time (BT), flight time (FT), body entry time (BET) in the water, and the times at 5, 15, and 25 m [7]. Several studies have investigated the influence of the adjustable kickplate on body positioning and start biomechanics, reporting significant differences between GS and KS, particularly regarding horizontal impulse and force production during the block phase [3,8]. Furthermore, the block time associated with the KS is reportedly shorter than that of the GS, primarily due to the horizontal reaction force produced by the hands and rear leg during the initial movement phase [9]. While the GS is typically linked to limited horizontal propulsion during the block departure phase, this aspect is considered disadvantageous in CS, as it prolongs the flight phase and increases the non-propulsive distance covered before water entry [9]. Although KS has been the subject of several investigations in CS, relatively few studies have focused specifically on its kinematic characteristics and detailed performance impacts at distances of 5, 10, and 15 m, respectively [2]. Based on Tor et al. (2014), male swimmers generally demonstrate a shorter reaction time, a greater horizontal take-off velocity, longer flight distances, and faster overall start times (5–15 m) compared to females [2].

The functional distinction between the front and rear legs during the KS arises from the inclined kickplate, which alters the lever arm mechanics and rear leg positioning [10]. By generating a significant horizontal impulse during the initial block phase, the rear leg contributes to the take-off horizontal velocity [11], which has been reported to account for 81% of the variance in start performance during the overwater phase [5]. Conversely, the front leg predominantly provides a vertical impulse, influencing vertical launch velocity and optimizing the water entry angle [12]. A distinctive feature of the KS is the simultaneous force application by both upper and lower limbs, creating a closed kinetic chain where arm and leg forces interact synergistically [12].

Despite the extensive body of research on the CS start, there remains a significant gap in the literature concerning the BFS start, especially since the World Underwater Federation (Confédération Mondiale des Activités Subaquatiques—CMAS) regulations still allow the use of the GS. To date, only two studies have examined start techniques in FS. Liashenko et al. (2019) [13] conducted a broader kinematic analysis, evaluating reaction time, horizontal flight velocity, and acceleration across CS, monofin swimming, and BFS. The findings revealed distinct velocity and acceleration profiles among athletes, indicating that start performance varies according to the swimming discipline and fin type [13]. Also, Stavrou and Voutselas (2018) [14] investigated variations in hand placement on the starting block during GS with a monofin but did not analyze other kinematic parameters [14]. These results emphasize the need for further biomechanical research into FS starts. The objective of this study is to experimentally investigate the biomechanical and performance differences between the GS and the KS in the BFS, by comparing all start performance variables and examining the relationships between the start and its associated variables. This investigation will be the first to examine these disparities with the leg positioned at either the front or the rear of the starting block.

## 2. Materials and Methods

### 2.1. Participants

The prediction of the required sample size (n) was conducted using G*Power 3.1.9.7 (Universität Düsseldorf Heinrich Heine University Düsseldorf, Düsseldorf, Germany) for Windows [15]. To detect significant differences between two groups (males and females) across three starting techniques—GS, KS with the right leg on the kickplate (KSR), and KS with the left leg on the kickplate (KSL)—a repeated measures ANOVA with within- and between-group interactions was conducted. A large effect size (f = 0.60) was observed. Power analysis indicated that a minimum sample size of 12 participants would be required to achieve a statistical power of approximately 96%. This would provide a modest probability of rejecting the null hypothesis under these conditions.

The present study involved thirteen (n = 13) national-level finswimmers (7 males and 6 females) who voluntarily participated in the experiment. The participants’ demographic and anthropometric characteristics were as follows (mean ± SD): age = 17.7 ± 2.1 years; height (H) = 1.75 ± 0.09 m; weight (W) = 69.6 ± 12.5 kg. Additionally, data on preferred leg placement on the kickplate and European shoe size (fin size—S) were recorded (Table 1).

Before the experimental procedures began, all participants and their parents and coaches were thoroughly informed about the study’s purpose and protocol. During data collection, all participants were healthy, injury-free, and well rested. The study protocol received approval from the Local Ethics Research Committee of the School of Physical Education and Sport Science at Serres, Aristotle University of Thessaloniki (ERC-007/2025, 14 March 2025), and was conducted in accordance with the Declaration of Helsinki. For underage participants, informed written consent was obtained from their legal guardians.

### 2.2. Experimental Procedure

Initial assessments included age, height, weight, and identifying each participant’s preferred leg placement on the kickplate of the starting block (Table 1). Height was measured using a portable stadiometer (Tanita HR001, Tanita Corporation, Tokyo, Japan), while weight was recorded with a digital scale (Tanita RD-545 HR, Tanita Corporation, Tokyo, Japan). The experimental trials took place in a 50 m indoor swimming pool, where the water temperature was kept between 25 and 26 °C and the ambient temperature ranged from 22 to 25 °C. Before the experimental procedure, all participants completed a standardized precompetition warmup, followed by a 10 min rest interval protocol, with the protocol outlined by Jimenez-Perez [16].

Each finswimmer completed two attempts for each of the three starting techniques: the first attempt served as a familiarization trial, while the second was used to collect performance data. The three starting conditions, GS, KSR, and KSL (Figure 1), were evaluated on separate days (a total of three testing days), with a 10 min rest period between the two attempts conducted each day.

The order of start techniques was randomized for each participant using a counterbalanced design to minimize potential order effects and systematic bias. All trials were conducted under simulated race conditions, closely replicating official competition settings in accordance with CMAS regulations.

### 2.3. Video Recording

A total of four cameras were utilized to record each fin swimmer’s start in a simultaneous manner. These cameras were meticulously synchronized using an LED light positioned directly in front of them. This LED light served as a standardized visual start signal for all the recording systems (Figure 2).

The first and second cameras (C1 and C2) were positioned at a height of 1.70 m above the surface of the pool deck and at a horizontal distance of 3 m on either side of the starting block. The third camera (C3) was positioned underwater at a depth of 0.90 m and a horizontal distance of 5 m from the side wall of the pool. A fourth camera (C4) was positioned at the side of the pool deck, 15 m from the starting block and at a height of 0.5 m (Figure 2).

The start signal was given by an electronic horn precisely synchronized with four LED lights. Each LED was positioned directly in front of one of the four cameras, providing a simultaneous visual cue. This setup allowed exact timing alignment between the start signal and the video recordings from all camera angles.

### 2.4. Recording Equipment

Data acquisition was performed using three cameras (GoPro Hero 9, San Mateo, CA, USA) and one underwater camera (GoPro Hero 12, San Mateo, CA, USA), all operating at a frame rate of 50 frames per second (fps). The cameras featured 23.6 megapixel image sensors and HyperSmooth 3.0 stabilization, capable of recording up to 240 fps at 1080p resolution, up to 60 fps at 4K, and a maximum resolution of 5K at 30 fps.

### 2.5. Parameter Analysis

The timings for the distances of 5 m, 15 m, and 25 m were recorded (T5GS, T5KSR, T5KSL, T15GS, T15KSR, T15KSL, T25GS, T25KSR, and T25KSL). The 5 m and 15 m times were recorded using cameras C1 and C3, respectively, measuring the interval from the LED flash (start signal) to the moment when the finswimmer’s snorkel crossed the corresponding virtual 5 m or 15 m line. The 25 m time was measured manually using handheld stopwatches at the point when the snorkel crossed the virtual 25 m line. Two experienced certified timekeepers operated independent handheld electronic stopwatches, and their recordings were averaged for analysis.

Video recordings were analyzed using Kinovea software (v. 2024.1), a validated tool for two-dimensional motion analysis widely used in biomechanical research [17]. Each parameter was extracted separately for the three start techniques (GS, KSR, and KSL), contributing to a comprehensive evaluation of start-phase performance in finswimming. Reaction time was defined as the interval between the start signal and the first observable movement (RTGS, RTKSR, and RTKSL). Block time was defined as the total time that the athlete remained in contact with the starting block after the start signal (BTGS, BTKSR, and BTKSL). Flight distance was defined as the horizontal distance from the block to the point where the swimmer’s fingertips touched the water (FDGS, FDKSR, and FDKSL). The flight time was defined as the interval between the moment when the swimmer’s fins lost contact with the starting block and the moment when their fingertips first touched the water (FTGS, FTKSR, and FTKSL). Additionally, the angle of body entry into the water was measured (BEAGS, BEAKSR, and BEAKSL). Body entry time was defined as the time from initial fingertip contact with the water surface until complete submersion of the body and fins (BETGS, BETKSR, and BETKSL). Diving depth was recorded at a distance of 5 m from the wall (DGS, DKSR, and DKSL). The distance covered during the underwater phase was recorded (UWGS, UWKSR, and UWKSL). Finally, the number of underwater dolphin kicks performed from entry to breakout was recorded (DKGS, DKKSR, and DKKSL).

A series of additional kinematic parameters were recorded, including joint angles, which were marked on anatomical landmarks corresponding to each joint. The following joint angles were measured: hip angle in GS (HAGS), front leg hip angles in KSR and KSL (FHAKSR and FHAKSL), back leg hip angles in KSR and KSL (BHAKSR and BHAKSL), knee angle in GS (KAGS), front leg knee angles in KSR and KSL (FKAKSR and FKAKSL), back leg knee angles in KSR and KSL (BKAKSR and BKAKSL), ankle angle in GS (AAGS) and front leg ankle angles in KSR and KSL (FAAKSR and FAAKSL), and back leg ankle angles in KSR and KSL (BAAKSR and BAAKSL). To ensure the validity of the findings, the kinematic parameters were analyzed using Kinovea by a single experienced performance analyst, in consultation with the head coach, who has extensive expertise and a background in competitive swimming.

### 2.6. Statistical Analysis

All data are presented as mean ± standard deviation (SD). Descriptive statistics were first calculated for all variables. The normality of distributions was assessed with the Shapiro–Wilk test. Homogeneity of covariance and sphericity were examined using Box’s test of equality of covariance matrices and Mauchly’s test, respectively. When the assumption of sphericity was violated, the Greenhouse–Geisser correction was applied. A two-way repeated measures analysis of variance (ANOVA) (start type × gender) was conducted to analyze the following variables: T5, T15, T25, RT, BT, FD, FT, BEA, BET, D, UW, and DK. Within- and between-subject effects were tested, and pairwise comparisons were performed using Bonferroni’s post hoc test in cases of significant differences. Additionally, paired-sample *t*-tests were used to compare HA, KA, and AA between the front and rear legs in KS. Effect sizes (ESs) were estimated using partial eta squared (η^2^_p_), with thresholds interpreted as small (≥0.01), medium (≥0.06), and large (≥0.14) [18]. Finally, correlations between the type of start (dependent variable) and start-related measures (independent variables) were examined using bivariate Pearson’s r. Correlation strength was interpreted as follows: r < 0.10 = negligible, 0.10–0.39 = weak, 0.40–0.69 = moderate, 0.70–0.89 = strong, ≥0.90 = very strong. All statistical analyses were performed with SPSS software (Version 25.0; IBM Corp., Armonk, NY, USA). The level of significance was set at α = 0.05.

## 3. Results

According to the Shapiro–Wilk test (for samples ≤ 30 participants), the data were normally distributed (*p* > 0.05). Thus, parametric analyses were followed for all examined variables. Additionally, according to Box’s and Mauchly’s tests, respectively, the data for the finswimmers’ start, and performance variables showed homogeneity and sphericity (*p* > 0.05). For FTGS, FTKSR, and FTKSL, which did not meet the sphericity criterion (*p* < 0.05), a Greenhouse–Geisser sphericity analysis was conducted. Moreover, there was no statistically significant difference between genders (*p* > 0.05) in any examined parameter. For this reason, the analysis was conducted for both genders combined for all parameters.

The analysis of the data indicated that the T5GS was significantly more extended than the T5KSR and T5KSL (1.79 ± 0.13 vs. 1.66 ± 0.14 and 1.68 ± 0.13 s, respectively, *p* < 0.001, η^2^_p_ = 0.6), the T15GS was significantly more extended than the T15KSR, T15KSL (6.0 ± 0.45 vs. 5.86 ± 0.40 and 5.88 ± 0.44 s, respectively, *p* < 0.001, η^2^_p_ = 0.5), and the T25GS was significantly more extended than the T25KSR and T25KSL (10.74 ± 0.74 vs. 10.64 ± 0.73 and 10.61 ± 0.71 s, respectively, *p* < 0.05, η^2^_p_ = 0.3). The BTGS was significantly more extended than the BTKSR and BTKSL (0.83 ± 0.04 vs. 0.71 ± 0.08 and 0.69 ± 0.06 s, respectively, *p* < 0.001, η^2^_p_ = 0.8). The FTGS was significantly shorter than the FTKSR and FTKSL (0.22 ± 0.07 vs. 0.26 ± 0.06 and 0.28 ± 0.06 s, respectively, *p* = 0.002, η^2^_p_ = 0.4). Apart from the parameters that showed significant differences, no statistically significant differences were observed among the remaining variables across the three starting conditions (Table 2).

Table 3 presents the results of the correlation analysis, revealing several significant relationships between performance outcomes and kinematic variables. Faster times at T5, T15, and T25 were positively associated with greater anthropometric parameters (H, W, and S), as well as with increased FDGS, FDKSR, FDKSL across the three start types, extended flight time in FTKSL, and shorter BETKSL. Additionally, shorter BTGS was linked to longer FDGS and FTGS, while shorter BTKSL was associated with longer FTKSL. Longer FDGS, FDKSR, and FDKSL were each correlated with longer FTGS, FTKSR, and FTKSL and smaller BEAGS, BEAKSR, and BEAKSL, respectively. Specifically, a longer FDKSL was associated with a faster BETKSL, and a longer FTKSL corresponded with a reduced BETKSL. Furthermore, a smaller BEAKSL was also linked to a faster BETKSL. Greater BEAKSR and BEAKSL were associated with superior DKSR and DKSL, respectively. Lastly, higher DKSR and DKSL values were positively correlated with extended UWKSR and UWKSL and were further associated with a greater number of DKKSR and DKKSL, respectively.

## 4. Discussion

In the present study, most of the results will be discussed in comparison with data from the literature on CS, as this is the first study conducted on BFS. Significant differences were observed in performance over the 5, 15, and 25 m distances in BFS when different starting techniques were employed. Specifically, times at 5, 15, and 25 m were significantly faster with the KSR and KSL starts compared to the GS. A comparable approach to the present study was identified in the work of Silveira et al. (2018), in which it was observed that front and rear KS resulted in faster performance times at 5 m compared to GS (1.90 ± 0.19 s and 1.89 ± 0.16 s vs. 1.98 ± 0.15 s, respectively), suggesting a potential advantage of this technique during the initial phase of the start [3]. Similar values to those in the present study were found in the study by Honda et al. (2012) (≅1.63 s) in KS with a different kickplate position [19]. Similar values to ours at 15 m after GS (5.70 ± 0.50 s) were found by Stavrou and Voutselas (2018) [14]. In contrast, the findings of Vantorre et al. (2010) in CS indicate an absence of disparities in the 15 m start time performance between the GS and TS (≅6.5 s), indicating that the different sport and the different type of starting block influence the times at 15 m [7]. Similar times were found for KS in CS at 15 m, as reported by Burkhardt et al. (2023) (≅7.0 s), where differences from changing from preferred to non-preferred leg position appeared to have a negative effect on starting performance, regardless of whether the stronger leg was in front or behind [8]. Also, Murrell and Dragunas (2012) report, in their introduction, that analysis of kinematic data in previous studies had shown no difference in start times to finger entry, 6.07 m, 10 m, or 15 m between GS and TS [6]. Another study, by Tor et al. (2014), examined the KS, where male swimmers achieved times at 5 m and 15 m of 1.47 ± 0.05 s and 6.12 ± 0.16 s, respectively, while female swimmers recorded 1.67 ± 0.08 s and 7.07 ± 0.28 s, respectively [2].

The RT should be as short as possible, while the movement phases in the block biomechanics of the swim start must last long enough to maximize the swimmer’s impulse to reach a high horizontal velocity [7,20]. In the present study, no statistically significant differences were observed in RT among the GS, KSR, and KSL techniques. These findings align with previous research in competitive swimming, where the start technique did not significantly affect RT. Vantorre et al. (2010) reported similar RT values between the GS (0.20 ± 0.03 s) and the traditional TS (0.20 ± 0.02 s) [7]. Also, according to the study by Blanksby et al. (2002), the type of start did not significantly affect RT [9]. The study revealed no statistically significant differences between the techniques. The pre-intervention measurements indicated a mean reaction time of 0.21 ± 0.05 s for the GS and 0.23 ± 0.04 s for the TS. Following the intervention, the techniques demonstrated improvements of approximately 0.19 s.

The BT was significantly briefer in the KSR and KSL compared to the GS, indicating a faster detachment of the finswimmer from the starting block. These findings align with those in CS, where the GS technique is also associated with prolonged BT. Lee et al. (2012) documented a shorter BT during the TS (0.79 ± 0.05 s) than the GS (0.84 ± 0.07 s) and attributed this difference to the location of the center of mass [21]. While they discussed the potential role of body positioning in this difference, it is important to note that the center of mass was not directly measured in our study. Compared to the traditional GS, the KS reduced the on-block time [22,23]. According to the study by Welcher et al. (2008), a statistically significant difference was observed in BT between the GS = 0.87 ± 0.05 s and the rear-weighted TS = 0.87 ± 0.08 s vs. the front-weighted TS = 0.80 ± 0.06 s, with the front-weighted TS demonstrating a faster block time compared to the other two start variations [24]. In contrast, in the study by Blanksby et al. (2002), the type of start did not significantly affect BT [9]. The results showed no statistically significant differences between techniques, as the pre-intervention measurements indicated a mean block time of 0.86 ± 0.07 s for the GS and 0.88 ± 0.08 s for the TS. Following the intervention, the techniques demonstrated improvements of approximately 0.83 s. Nevertheless, evidence suggests that KS techniques facilitate more immediate acceleration, possibly due to the shorter BT and the more efficient execution of the start [21].

The FD remained comparable between the GS vs. KRS vs. KSL while the GS exhibited a shorter FT compared to the KSR and KSL. Regarding FD, the study by Jorgić et al. (2010) revealed that the FD of the GS (3.21 ± 0.17 m) is approximately 0.23 m greater than that of the TS (2.98 ± 0.13 m) [25]. Studies by Tor et al. (2015) and Honda et al. (2012) measured similar FDs (≅2.96 m and 2.74, respectively) [19,26] to those in our study. Our results differed significantly from those of Stavrou and Voutselas (2018) (130.3 ± 9.1 cm) [14], probably due to the measurement method. The shorter FT of the GS is consistent with previous studies on conventional swimming starts, which have shown that TS is linked to longer FT due to differences in push-off angles during the take-off phase. Cicenia et al. (2019) investigated FT across three different kick plate positions and found no statistically significant differences between groups, with the FT being approximately 0.30 s, similar to our own findings [27]. This phenomenon may be explained by a higher initial horizontal take-off velocity in the GS, possibly due to the symmetrical foot placement that allows for a more powerful push-off from the block [10]. Matúš et al. (2021) support the idea that asymmetrical stances often result in a more vertical and less linear push-off, leading to longer FT without increasing FD [4].

Regarding the BEA, no statistically significant differences were observed between the GS, KSR, and KSL. The literature does not provide clear data directly comparing the entry angle between the GS and the KS. However, it is considered that the flight trajectory followed by the athletes may influence the angle of entry. Jorgić et al. (2010) observed that the BEA in the GS (33.33 ± 5.13°) was approximately 4 degrees greater than in the TS (29.33 ± 5.51°) [25]. Their study found BEAs significantly smaller than those observed in our study, but similarly, no statistically significant difference in BEA measurement was found between GS and TS. Similar, in the study conducted by Silveira et al. (2018), the BEA was found to be 36.9° ± 5.1° for the GS, 37.3° ± 3.6° for the Front TS, 37.5° ± 4.1° for the Rear TS, 37.8° ± 4.0° for the Front KS, and 36.8° ± 5.0° for the Rear KS [3]. Although slight variations were observed between techniques, these differences were not statistically significant, suggesting that BEA alone may not serve as a key differentiating factor across start variations. Furthermore, Lee et al. (2012) conducted a study that examined the BEA in both GS and TS techniques [21]. The study concluded that there was no statistically significant difference between them in terms of entry angles of 37.6 ± 5.9° for the GS and 40.9 ± 3.9° for the TS. In the study by Thanopoulos et al. (2012), no statistically significant differences in BEA were observed between the GS and TS techniques for either male (44.22° ± 5.58° vs. 43.85° ± 4.48°) or female swimmers (45.18° ± 4.02° vs. 44.79° ± 4.00°), and no gender difference was observed, consistent with the findings of the present study [28]. Overall, the findings indicate that BEA remains relatively consistent across start techniques and genders, suggesting that athletes can adapt both starting styles for effective body entry into the water.

Regarding the ΒΕΤ, no statistically significant differences were observed among the three starting techniques examined in the present study. Guimaraes and Hay (1985) and Hay (1988) concluded that BET was more important to the start phase than either block time or flight time (explaining 95% of the variance in the starting time for r = 0.97) [29,30,31]. Based on the results presented by Silveira et al. (2018), the BET was reported as 0.35 ± 0.04 s for the GS, 0.35 ± 0.05 s for the Front TS, 0.34 ± 0.04 s for the Rear TS, 0.35 ± 0.04 s for the Front KS, and 0.33 ± 0.04 s for the Rear KS [3]. Also, as reported by Vantorre et al. (2010), the BET into the water did not significantly differ based on the start technique, with entry times recorded as 0.28 ± 0.04 s for the GS and 0.27 ± 0.12 s for the TS [7]. These findings are very close to ours (note that our measurements included the time it took for the fins to be submerged); although slight variations were observed among the different start techniques, these differences were not statistically significant, indicating that BET alone may not be a critical factor in distinguishing start performance across start styles [3,7].

No statistically significant differences were documented in the D, in the UW, or in the DK. The study by Matúš and Kandráč (2020) found that maximum D upon water entry varied (0.75 to 0.92 m) based on the swimmer’s starting position and the OSB12 kick plate setting [32]. A recent study by Matúš et al. (2021) found that the maximum underwater depth was 0.90 m [4]. This aligns with recommendations from Tor et al. (2015), who researched optimizing underwater trajectories [5]. They suggest that swimmers should aim for a maximum depth of about 0.92 m to reduce velocity loss during the underwater phase. UW is more likely influenced by post-entry technique, body alignment, and propulsion efficiency during the glide and dolphin kick phases than by the initial start method. In accordance with the findings and recommendations set out by Tor et al. (2015), it is recommended that swimmers extend UW and delay the initiation of the first undulatory kick at approximately 6.6 m, with the aim of minimizing the velocity lost during the underwater phase and enhancing performance [26]. According to Veiga et al. (2014, 2022), the number of DK exhibited significant variation depending on the experimental condition [33,34]. Calculations indicated that the number of underwater DK at 15 m ranged from 8 to 12 kicks in the underwater segments of the start and turns.

No statistically significant differences were found between KSR and KSL in FHA, BHA, FKA, BKA, FAA, or BAA. In the present study, the level of HA was found to be marginally elevated in GS relative to KS, with the range of values observed aligning closely with the extant literature on CS for GS and TS, as reported in the study by Vantorre et al. (2011) (26–30°) [35]. In contrast, the studies conducted by Matúš and Kandráč (2020) and Matúš et al. (2021) revealed that the range of HA in KS varied from 41° to 48° [4,32]. With regard to the KA in GS, the present study’s findings exhibited a considerable divergence from those reported by Stavrou and Voutselas (2018), who determined a value of 107.1 ± 8.9° [14]. In consideration of the KA in KS, the findings of the study of Matúš and Kandráč et al. (2020) in CS demonstrate a result relevant to those obtained in our study [32]. The study’s objective was to compare starts on the OSB12 starting block in different kickplate positions and with neutral-weighted and rear-weighted starts. The study’s findings indicated that the lowest and highest front and rear knee angles (124.8° ± 1.9° to 132.7° ± 1.8°, and 76.7° ± 1.5° to 96.0° ± 1.6°, respectively) were observed. The KAs at the KS that were observed in the present study appear to be within the appropriate range. These observations are corroborated by the findings of Matúš et al. (2021) in CS, who posit that the angles of the front and rear knees are 135–140° and 75–85°, respectively [4].

The correlation analysis revealed several statistically significant relationships between performance indicators (times at 5, 15, and 25 m) and both anthropometric parameters and kinematic start components. Faster T5, T15, and T25 times were correlated with greater H, W, and S, according to previous research [29], and increased FD across all start types was in line in the study by Ruschel et al. (2007) regarding GS (r = −0.482) [36]. The extended FTKSL was in line with results from the study by Vantorre et al. (2010) (r = −0.31, *p* = 0.05) [7]; however, Ruschel et al. (2007) reported that there was no correlation between FT and T15 (r = 0.039) [36], and they reported a shorter BETKSL. The findings of the present study suggest that swimmers with greater anthropometric characteristics have the potential to achieve enhanced performance, which is associated with extended FT and shorter BET during KS executed with the left leg positioned on the kickplate. In our study, there was a lack of correlation of T5, Τ15, Τ25 with KA or BT, which had been observed in previous studies [4,7,24]; however, the findings were consistent with those reported by Ruschel et al. (2007), who observed a correlation between T15 and BT (r = −0.115) in GS [36].

A shorter BTGS was associated with longer FDGS and FTGS. Similarly, a shorter BTKSL was linked to a longer FTKSL, indicating a trade-off between time spent on the block and the effectiveness of the flight phase. Vantorre et al. (2010) posit that the block phase must be sufficiently prolonged to optimize the impulse, thereby ensuring that the swimmer exits the block with the greatest possible horizontal velocity [37].

Also, correlation analysis showed that lengthier FD was significantly associated with extended FT and smaller BEA in all start types. Taladriz et al. (2017) observed that, although the KS produced a higher deceleration rate than the GS, this phenomenon resulted in the swimmers experiencing a lower acceleration rate when their feet left the block [23]. This finding suggests that enhancing take-off capabilities may enable augmented horizontal displacement in the air without compromising entry efficiency. Specifically, a greater FDKSL was associated with a more rapid BETKSL, and a prolonged FTKSL was correlated with a diminished BETKSL. In accordance with the assertions put forward by Vantorre et al. (2014), the generation of sufficient momentum is imperative in ensuring entry into the water at a considerable velocity [30].

A smaller BEAKSL was found to be associated with a faster BETKSL. Vantorre et al. (2014) report that in order to manage the angular momentum generated during the block phase, swimmers have the capacity to make a start with a reduced angular momentum [30], indicating that swimmers who enter the water at a shallower angle may do so more expeditiously. In contrast, a positive correlation was observed between larger BEAKSR and BEAKSL and more DKSR and DKSL, respectively. This finding suggests that, in the case of KS, the larger the BEA, the greater the body’s emergence, thereby enhancing the efficacy of subsequent dolphin kicks. As asserted by Tor et al. (2015), the act of entering the water at a flatter angle can be counteracted if the swimmer fails to sustain velocity during the underwater phase [5]. This is attributable to an augmented amount of resistance being exerted on the swimmer. Finally, deeper DKSR and DKSL were strongly correlated with extended UWKSR and UWKSL and with a greater number of DKKSR and DKKSL, respectively. These findings suggest that swimmers capable of a deeper underwater trajectory generating more dolphin kicks can sustain underwater propulsion for longer durations [38].

In contrast to previous studies, our research found no significant correlations between body angles and any performance or kinematic variables. Slawson et al. (2012) reported strong positive correlations between rear knee angle and peak force in both horizontal (r = 0.701) and vertical directions (r = 0.688), as well as a moderate correlation between rear knee angle and peak horizontal force (r = 0.511) [10]. Similarly, Matuš et al. (2021) identified significant associations between start posture, particularly head, ankle, knee, and shoulder positions, and early race performance [4]. Notably, the front ankle angle correlated with block time (r = 0.63), while back knee (r = 0.56) and front ankle angles (r = 0.51) were also linked to 5 m time.

Based on the results and correlation analysis presented, we can summarize that the KS, particularly the KSL, appears to be the most effective and superior technique for finswimmers and consistently achieved faster times at 5 m, 15 m, and 25 m [39]. This indicates a more powerful and efficient start phase in short- and mid-range distances. Shorter BTKSL were observed in KSL, indicating that a faster transition of the block means more time profit. Longer FDKSL and longer FTKSL were positively correlated with performance. KSL allowed swimmers to travel further and stay airborne longer, improving the water entry angle and speed. Higher BEAKSL correlated with better DKSL and UW. This makes the KSL more advantageous for maximizing underwater efficiency before breaking out. Even though the majority of the finswimmers (11 out of 13) indicated that their right foot was both strong and preferable and placed it on the kickplate, it has been demonstrated that when it is positioned at the edge in front, it appears to enhance the start. In the study in [40], there was a similar preference with the left leg most preferred as the rear foot regularly used during a kick start (six vs. four swimmers), thereby underscoring the significance of the left leg positioned on the kickplate and the right at the front edge of the block as the preferable and strong leg.

The limitations of this study include the small sample size and the quantification and acknowledgment of potential measurement errors arising from video analysis, such as cameras angle, image linearity, and calibration. Although this is a common challenge in applied sport research, given the difficulty of recruiting larger samples of national-level finswimmers, the findings should be interpreted with some caution. Another limitation was the inability to accurately determine the position of the body’s center of mass on the starting block and the precise placement of the kickplate. Access to such data would have allowed for more comprehensive conclusions to be drawn. Nevertheless, the present findings contribute to a more precise determination of the optimal start for finswimmers.

### Recommendations for Coaches and Bi-Finswimmers

We recommend prioritizing KS over GS in the case of CMAS permitting this start. It is effective especially in sprint events, where start phases heavily impact the overall race time. We recommend emphasizing KSL, which yielded the fastest times at 5 m, 15 m, and 25 m and showed significant correlations with powerful underwater phases, and developing explosive strength and RT to reduce block time. We also recommend improving the effectiveness of DK and ensuring that the permissible UW is fully utilized and assessing the swimmer’s preferred leg to decide between KSR and KSL. Furthermore, it is a good idea to train both sides in younger swimmers to promote bilateral skill development.

## 5. Conclusions

The study’s findings indicate that the kick start technique generally provides superior performance advantages over the grab start, particularly in terms of block time, in Bi-Finswimming. These results provide preliminary evidence that the kick start could be considered for future discussion regarding its potential legalization by CMAS, although further research with larger and more diverse samples is needed before any regulatory recommendations are made. However, the performance of the start is multifactorial and is profoundly influenced by anthropometric variables, flight kinematics, block exit dynamics, and underwater kicking efficiency. Nevertheless, individualized assessments remain paramount, as certain athletes may demonstrate superior performance with the preferable start based on biomechanics and comfort.

## Figures and Tables

**Figure 1 jfmk-10-00384-f001:**
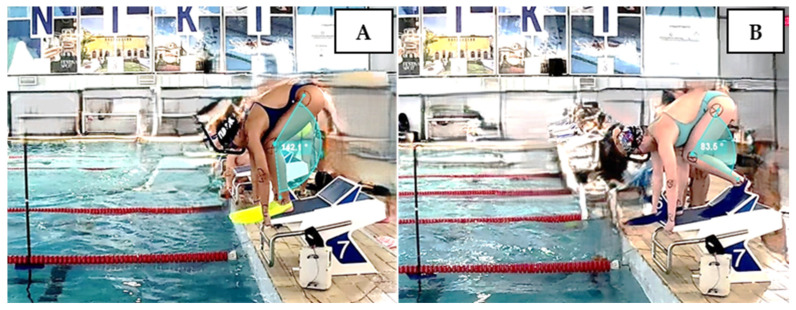
Grab start (**A**) and kick start (**B**) with the left foot on the kickplate.

**Figure 2 jfmk-10-00384-f002:**
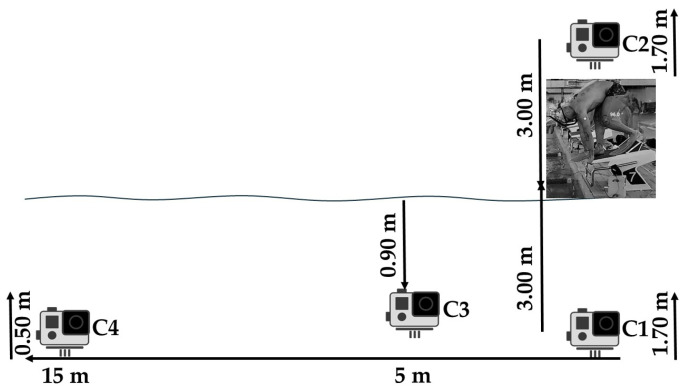
Locations of four fully synchronized cameras relative to the starting block edge and water surface.

**Table 1 jfmk-10-00384-t001:** The demographic and anthropometric characteristics of the participants are presented.

Athlete	Gender	Age (y)	Height (m)	Weight (kg)	Preferred Leg on the Kickplate	Size of Fins
1	M	17.7	1.88	85.3	R	44
2	M	17.3	1.75	65.6	R	41
3	M	18.4	1.70	69.4	R	41
4	M	15.7	1.83	73.2	R	41
5	M	19.5	1.88	87.2	R	44
6	M	16.6	1.83	83.5	R	44
7	M	23.4	1.77	86.4	L	41
8	F	17.2	1.70	67.2	R	38
9	F	16.0	1.70	58.6	R	38
10	F	15.5	1.57	50.7	R	38
11	F	16.9	1.69	58.3	R	38
12	F	16.7	1.71	62.1	R	38
13	F	19.2	1.68	57.4	L	38

y, years; m, meters; kg, kilograms; M, male; F, female; R, right leg; L, left leg.

**Table 2 jfmk-10-00384-t002:** The mean ± SD is used to show the values for times, lengths, and angles in the present study.

Parameter /// Start Type	GS	KSR	KSL	η^2^_p_
T5—Time 5 m (s)	1.79 ± 0.13 **	1.66 ± 0.14	1.68 ± 0.13	0.6
T15—Time 15 m (s)	6.00 ± 0.45 **	5.86 ± 0.40	5.88 ± 0.44	0.5
T25—Time 25 m (s)	10.74 ± 0.74 *	10.64 ± 0.73	10.61 ± 0.71	0.2
RT—Reaction time (s)	0.21 ± 0.02	0.22 ± 0.02	0.20 ± 0.03	0.15
BT—Block time (s)	0.83 ± 0.04 **	0.71 ± 0.08	0.69 ± 0.06	0.8
FD—Flight distance (m)	2.9 ± 0.3	2.9 ± 0.3	3.0 ± 0.4	0.1
FT—Flight time (s)	0.22 ± 0.07 **	0.26 ± 0.06	0.28 ± 0.06	0.4
BEA—Angle of body entry (^o^)	41 ± 8	42 ± 5	39 ± 6	0.09
BET—Body entry time (s)	0.37 ± 0.02	0.37 ± 0.03	0.37 ± 0.05	0.07
D—Diving depth (m)	0.92 ± 0.12	0.93 ± 0.27	0.92 ± 0.30	0.06
UW—Underwater distance (m)	13.4 ± 0.8	13.2 ± 1.1	13.3 ± 0.9	0.06
DK—Dolphin kicks (n)	8 ± 1	8 ± 1	8 ± 1	0.2
HA—Hip angle (^o^)	29 ± 4			
FHA—Front hip angle (^o^)		26 ± 4	24 ± 3	0.4
BHA—Back hip angle (^o^)		37 ± 4	40 ± 5	0.04
KA—Knee angle (^o^)	141 ± 9			
FKA—Front knee angle (^o^)		144 ± 9	141 ± 9	0.08
BKA—Back knee angle (^o^)		85 ± 10	87 ± 13	0.34
AA—Ankle angles (^o^)	97 ± 7			
FAA—Front ankle angles (^o^)		103 ± 8	101 ± 7	0.09
BAA—Back ankle angles (^o^)		77 ± 8	72 ± 10	0.2

GS, grab start; KSR, the right leg on the kickplate; KSL, the left leg on the kickplate; s, seconds; m, meters; ^o^, degrees; ** *p* < 0.001; * *p* < 0.05. Asterisks denote significant differences from the other two starting techniques.

**Table 3 jfmk-10-00384-t003:** Notable correlations among the examined parameters. It is important to note that the correlations refer to the parameters belonging to each start type.

Parameter/Start Type	GS	KSR	KSL
	Pearson’s *r*/*p*	Pearson’s *r*/*p*	Pearson’s *r*/*p*
T5—H	−0.720/0.006		−0.716/0.006
T15—H	−0.675/0.011	−0.587/0.035	−0.618/0.024
T25—H	−0.738/0.004	−0.682/0.010	−0.723/0.005
T5—W	−0.798/0.001		−0.680/0.011
T15—W	−0.792/0.001	−0.762/0.002	−0.751/0.003
T25—W	−0.838/0.000	−0.774/0.002	−0.773/0.002
T5—S	−0.704/0.007		−0.756/0.003
T15—S	−0.741/0.004	−0.726/0.005	−0.711/0.006
T25—S	−0.813/0.001	−0.803/0.001	−0.828/0.000
T5—FD	−0.738/0.004	−0.768/0.002	−0.798/0.001
T15—FD	−0.667/0.013	−0.855/0.000	−0.769/0.002
T25—FD	−0.670/0.012	−0.798/0.001	−0.806/0.001
T5—FT			−0.707/0.007
T15—FT			−0.663/0.013
T25—FT			−0.691/0.009
T5—BET			0.652/0.016
T15—BET			0.696/0.008
T25—BET			0.690/0.009
BT—FD	−0.616/0.025		
BT—FT	−0.694/0.008		−0.656/0.015
FD—FT	0.751/0.003	0.638/0.019	0.745/0.003
FD—BEA	−0.579/0.038	−0.638/0.019	−0.691/0.009
FD—BET			−0.727/0.005
FT—BET			−0.773/0.002
BEA—BET			0.652/0.016
BEA—D		0.709/0.007	0.604/0.029
D—UW		0.692/0.009	0.587/0.035
D—DK		0.568/0.043	0.701/0.008

GS, grab start; KSR, the right leg on the kickplate; KSL, the left leg on the kickplate; T5, T15, T25, times at 5, 15, and 25 m; H, height; W, weight; S, European fin size; FD, flight distance; FT, flight time; BET, time of the body’s entry into the water; BT, time on the starting block; BEA, angle of the body’s entry; D, diving depth; UW, underwater glide; DK, dolphin kicks.

## Data Availability

The data that supports the findings of this study are available upon request from the corresponding author.

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
