# Peer review of "Kinematic Comparison of Different Types of Start Technique in Bi-Finswimming"

_jfmk, 2025, doi:10.3390/jfmk10040384_

Round 1

Reviewer 1 Report

Comments and Suggestions for Authors

Title

add the type of comparison, by example kinematic comparison

Abstract

what is OSB11  and CMAS? idem introduction, please explain it

Mention the statistical analysis 

Introduction

please add the justification for compare genders

line 49-50 explain the track start and kick start and her differences

line 52: You dont use FT, BT, RT and BET, the is not necessary use it

Methods

Figure 1: explain in the caption what is image is GB or KS, by example, left image is GB and right is KS or add letters in the image and mention in the caption

Participants

What analysis was used in the Gpower (test family and statistical test)? what effect size is mentioned ? f of cohen ?

statistical analysis

delete this: were analyzed via Syntax

This subsection is very confuse, i suggest reorder

why correlations were performed and what is the importance of this analysis ? not is mentioned in the objetive of study

Results

Make a table for each factor analyzed and add the effect sizes.

Discussion

I believe that the methods used could have limitations compared to gold standard instruments, as well as the small sample size.

Author Response

REVIEWER 1

We really appreciate the positive and encouraging feedback from the reviewer.

Title

Comments 1: add the type of comparison, by example kinematic comparison

Response 1: Thank you for your suggestion. The title has changed to “Kinematic comparison of Different Type Start Techniques in Bi-Finswimming.

Abstract

Comments 2: what is OSB11 and CMAS? idem introduction, please explain it DONE

Response 2: Thank you for your comments. OSB11 is a well-known type of swimming Block, equipped with an adjustable, inclined rear kickplate.  CMAS is the World Underwater Federation (Confédération Mondiale Des Activités Subaquatiques). Explanations added.

Comments 3: Mention the statistical analysis

Response 3: The statistical analysis was incorporated into the abstract.

“A two-way repeated measures ANOVA (start type × gender) was conducted to analyze performance metrics, and a paired-sample t-test assessed differences in joint angles. Also, correlations between dependent (type of start) and independent variables (start-examined variables) were examined through bivariate Pearson's r analysis.”

Introduction

Comments 4: please add the justification for compare genders

Response 4: The justification was incorporated into the Introduction.

“Based on Tor et al. (2014), male swimmers generally demonstrate shorter reaction time, greater horizontal take-off velocity, longer flight distances, and faster overall start times (5–15 m) compared to females [2].”

Comments 5: line 49-50 explain the track start and kick start and her differences

Response 5: The explanation was incorporated into the Introduction.

“Since the introduction of the Omega OSB11 starting platform, equipped with an in-clined kickplate at the rear, considerable research in competitive swimming (CS) has focused on analyzing and comparing starting techniques. Particular attention has been given to the differences among the traditional grab start (GS), the track start (TS), which involves a split stance on the block with one foot forward gripping the front edge and the other positioned backward for leverage, and the kick start (KS), where the rear foot is placed on the inclined kickplate [3, 4].”

Comments 6: line 52: You dont use FT, BT, RT and BET, the is not necessary use it

Response 6: Dear reviewer, you will see that the shortcuts were also used below, just like in methodology and the statistics section.

Methods

Comments 7: Figure 1: explain in the caption what is image is GB or KS, by example, left image is GB and right is KS or add letters in the image and mention in the caption

Response 7: The explanation was incorporated into the figure’s caption.

“Figure 1. Grab Start (A) and Kick Start (B) with the left foot on the kickplate.”

Participants

Comments 8: What analysis was used in the Gpower (test family and statistical test)? what effect size is mentioned ? f of cohen ?

Response 8: We conducted repeated-measures F-tests for both within- and between-subjects interactions. The effect size was large (f = 0.60), according to Cohen’s criteria.

statistical analysis

Comments 9: delete this: were analyzed via Syntax

Response 9: The phrase underwent a modification in the statistical analysis section.

“Possible statistically significant differences were examined using pairwise comparisons between groups with Bonferroni’s post hoc test.”

Comments 10: This subsection is very confuse, i suggest reorder

Response 10: Dear reviewer, the subsection of statistical analysis has been reordered as suggested.

“All data are presented as mean ± standard deviation (SD). Descriptive statistics were first calculated for all variables. The normality of distributions was assessed with the Shapiro–Wilk test. Homogeneity of covariance and sphericity were examined using Box’s test of equality of covariance matrices and Mauchly’s test, respectively. When the assumption of sphericity was violated, the Greenhouse–Geisser correction was ap-plied. A two-way repeated-measures analysis of variance (ANOVA) (start type × gen-der) was conducted to analyze the following variables: T5, T15, T25, RT, BT, FD, FT, BEA, BET, D, UW, and DK. Within- and between-subject effects were tested, and pair-wise comparisons were performed using Bonferroni’s post hoc test in cases of signifi-cant differences. Additionally, paired-sample t-tests were used to compare HA, KA, and AA between the front and rear legs in KS. Effect sizes (ES) were estimated using Pillai’s partial eta squared (η²p), with thresholds interpreted as small (≥0.01), medium (≥0.06), and large (≥0.14) [18]. Finally, correlations between the type of start (depend-ent variable) and start-related measures (independent variables) were examined using bivariate Pearson’s r. All statistical analyses were performed with SPSS software (Ver-sion 25.0; IBM Corp., Armonk, NY, USA). The level of significance was set at α = 0.05.”

Comments 11: why correlations were performed and what is the importance of this analysis ? not is mentioned in the objetive of study

Response 11: The phrase underwent a modification at the end of introduction section.

“The objective of this study is to experimentally investigate the biomechanical and performance differences between the GS and the KS in the BFS, by comparing all start performance variables and examining the relationships between the start and its associated variables.”

Results

Comments 12: Make a table for each factor analyzed and add the effect sizes.

Response 12: Thank you for the suggestion. We included in Table 2 the effect sizes

Discussion

Comments 13: I believe that the methods used could have limitations compared to gold standard instruments, as well as the small sample size.

Response 13: The limitations were modified at the end of discussion section

«The limitations of this study include the small sample size, the quantification and acknowledgment of potential measurement errors arising from video analysis, such as cameras angle, image linearity, and calibration. Another limitation was the inability to accurately determine the position of the body’s center of mass on the starting block and the precise placement of the kickplate. Access to such data would have allowed for more comprehensive conclusions to be drawn. Nevertheless, the present findings contribute to a more precise determination of the optimal start for finswimmers.»

Reviewer 2 Report

Comments and Suggestions for Authors

Dear Editor, thank you for the opportunity to review this interesting work comparing three different starting techniques in Bi-Finswimming. Congratulations to the authors for their work. Overall, the study is well presented; however, in my opinion, some points could be considered to improve the manuscript before publication.

Specific comments.

Introduction: The introduction provides an overview of the literature and the problem to be addressed. However, in my opinion, the authors could have clearly stated the study's hypothesis in the last part of the introduction.

Methods: In my opinion, the video analysis procedures could be better described in this section, particularly the use and positioning of markers (visible in Figure 1) in relation to the relevant anatomical landmarks. How many operators worked on the video analysis on Kinovea? Has the measurement error been estimated?
The captions for Figures 1 and 2 could be better described.

The statistical analysis is theoretically correct. Although the data follows a normal distribution, given the small sample size and the variability of the data (Table 2), the use of the median and non-parametric statistical tools could be more robust as they are less affected by outliers, and I wondered if they were taken into consideration by the authors to confirm the results discussed.

Discussions and conclusion: In discussions, subparagraphs are not used if there is only one. In any case, I would place more emphasis on the limitations of the study, quantifying and highlighting the measurement errors that may result from video analysis (camera angle, image linearity, calibration, and operator errors dependent on the analysis on Kinovea). In this regard, artificial intelligence techniques and wearable devices could reduce measurement bias. For example, see: “Annino, Giuseppe, et al. "Assessing sports performances using an artificial intelligence-driven system." 2023 IEEE International Workshop on Sport, Technology and Research (STAR). IEEE, 2023.”; “Chern, Yinq-Rong, et al. "A butterfly stroke swimming recording and performance analysis system based on computer vision and machine learning." Measurement 251 (2025): 117171.”

The references are adequate.

Following these changes, in my opinion, the paper deserves to be considered for publication.

Author Response

REVIEWER 2

Dear Editor, thank you for the opportunity to review this interesting work comparing three different starting techniques in Bi-Finswimming. Congratulations to the authors for their work. Overall, the study is well presented; however, in my opinion, some points could be considered to improve the manuscript before publication.

Response: We would like to express our sincere gratitude to the reviewer for their positive and encouraging feedback.

Specific comments.

Comments 1: Introduction: The introduction provides an overview of the literature and the problem to be addressed. However, in my opinion, the authors could have clearly stated the study's hypothesis in the last part of the introduction.

Response 1: The phrase underwent a modification at the end of introduction section.

“The objective of this study is to experimentally investigate the biomechanical and performance differences between the GS and the KS in the BFS, by comparing all start performance variables and examining the relationships between the start and its associated variables.”

Comments 2: Methods: In my opinion, the video analysis procedures could be better described in this section, particularly the use and positioning of markers (visible in Figure 1) in relation to the relevant anatomical landmarks. How many operators worked on the video analysis on Kinovea? Has the measurement error been estimated?

Response 2: The phrase underwent a modification at the end of Parameters Analysis section.

“To ensure the validity of the findings, the kinematic parameters were analyzed using Kinovea by a single experienced performance analyst, in consultation with the head coach, who has extensive expertise and a background in competitive swimming.”

No measurement error has been estimated

Comments 3: The captions for Figures 1 and 2 could be better described.

Response 3: The figure's caption has undergone a modification.

“Figure 1. Grab Start (A) and Kick Start (B) with the left foot on the kickplate.”

Figure 2. Locations of four fully synchronized cameras relative to the starting block edge and water surface.”

Comments 4: The statistical analysis is theoretically correct. Although the data follows a normal distribution, given the small sample size and the variability of the data (Table 2), the use of the median and non-parametric statistical tools could be more robust as they are less affected by outliers, and I wondered if they were taken into consideration by the authors to confirm the results discussed.

Response 4: Dear Reviewer, we did not use the median or non-parametric statistical methods. Our discussion of the results is based on the findings from the statistical analyses that were actually applied.

Comments 5: Discussions and conclusion: In discussions, subparagraphs are not used if there is only one. In any case, I would place more emphasis on the limitations of the study, quantifying and highlighting the measurement errors that may result from video analysis (camera angle, image linearity, calibration, and operator errors dependent on the analysis on Kinovea). In this regard, artificial intelligence techniques and wearable devices could reduce measurement bias. For example, see: “Annino, Giuseppe, et al. "Assessing sports performances using an artificial intelligence-driven system." 2023 IEEE International Workshop on Sport, Technology and Research (STAR). IEEE, 2023.”; “Chern, Yinq-Rong, et al. "A butterfly stroke swimming recording and performance analysis system based on computer vision and machine learning." Measurement 251 (2025): 117171.”

Response 5: Dear reviewer, a) we agree with your view, but we thought that paragraphs in the discussion would help with a better reading of the text, b) we also agree with the view, AI techniques and mobile devices could reduce measurement errors, c) we emphasized the limitations you suggest.

The paragraph was added at the end of the discussion

“The limitations of this study include the small sample size, the quantification and acknowledgment of potential measurement errors arising from video analysis, such as cameras angle, image linearity, and calibration. Another limitation was the inability to accurately determine the position of the body’s center of mass on the starting block and the precise placement of the kickplate. Access to such data would have allowed for more comprehensive conclusions to be drawn. Nevertheless, the present findings contribute to a more precise determination of the optimal start for finswimmers.”

Comments 6: The references are adequate.

Response 6: Dear reviewer, thank you for your comment

Following these changes, in my opinion, the paper deserves to be considered for publication.

Response: Dear reviewer, thank you for your comment

Reviewer 3 Report

Comments and Suggestions for Authors

This paper covers an original and timely question: how different start techniques (Grab vs. Kick Start, with both leg placements) affect performance in Bi-Fin swimming. Since most of what we know comes from conventional swimming, bringing this discussion into BFS is genuinely valuable. The study is well motivated and has practical implications for both athletes and coaches, especially with the suggestion that Kick Start could be legalized in competition.

That said, I think the manuscript still needs quite a bit of work before it is ready. The introduction, while comprehensive, leans very heavily on the swimming literature and repeats points that are already well established there. It would be stronger if it focused more directly on what makes BFS unique (snorkel, fins, underwater dynamics) and why the findings may not align perfectly with conventional swimming research.

Methodologically, there are some concerns. The statistical power analysis shows that the study is underpowered (only ~63%), and while the sample size is reasonable for an applied sport context, this limitation should be clearly and explicitly acknowledged. The use of manual stopwatches for the 25 m times is problematic, given that the other distances were video-based — it introduces unnecessary measurement error. Finally, the way “preferred” versus “dominant” leg was defined seems too simplistic. A more objective assessment would have made the comparisons between KSR and KSL more convincing. Please clarify and justify this point.

The results are presented clearly; the tables are thorough, but the number of variables analyzed is quite large relative to the sample size. This makes some of the correlation findings look more exploratory than definitive. I also felt that some of the interpretations in the discussion went beyond what the data can actually support.

For example, explanations involving the center of mass, which was not measured. The discussion could be tightened up to focus on the genuinely new findings: Kick Start being faster than Grab Start in BFS, and the subtle differences between leg positions. Please focus on the main objectives/data of the study.

The practical recommendation — that Kick Start should be legalized — is bold and potentially important, but it would be better framed as preliminary evidence. With such a small and specific sample, it is hard to make such a broad regulatory claim. Still, the study makes a strong case that this discussion needs to happen.

In terms of writing, the paper is readable but dense. Some sections (especially the introduction and discussion) are overly long and would benefit from language editing to improve clarity and flow. The authors could streamline and condense the wording in these sections.

Overall, I see this as a relevant and potentially useful contribution, but in its current form it feels too speculative and somewhat overstated. With a more cautious interpretation, a sharper focus on what is truly new, and a clearer acknowledgment of the limitations, it could be a solid paper.

Author Response

REVIEWER 3

This paper covers an original and timely question: how different start techniques (Grab vs. Kick Start, with both leg placements) affect performance in Bi-Fin swimming. Since most of what we know comes from conventional swimming, bringing this discussion into BFS is genuinely valuable. The study is well motivated and has practical implications for both athletes and coaches, especially with the suggestion that Kick Start could be legalized in competition.

Response: We would like to express our sincere gratitude to the reviewer for their positive and encouraging feedback.

Comments 1: That said, I think the manuscript still needs quite a bit of work before it is ready. The introduction, while comprehensive, leans very heavily on the swimming literature and repeats points that are already well established there. It would be stronger if it focused more directly on what makes BFS unique (snorkel, fins, underwater dynamics) and why the findings may not align perfectly with conventional swimming research.

Response 1: We thank the reviewer for this constructive comment. We fully agree that BFS includes unique features such as snorkel use, fins, and underwater dynamics. However, the scope of the present study was intentionally limited to the start phase, with a specific focus on comparing two starting techniques performed primarily on the platform. To address the reviewer’s concern, we have revised the introduction by adding a paragraph to clarify our focus and to emphasize that our findings relate exclusively to the start and not to the full complexity of BFS performance.

“Although BFS performance is influenced by unique factors such as snorkel use, fins, and underwater dynamics, the present study does not aim to address the entirety of these elements. Instead, our focus is restricted to the start phase, as the effectiveness of the starting technique plays a decisive role in competitive outcomes. Previous research in CS has demonstrated that differences in start technique can significantly impact performance; however, it remains unclear whether similar patterns apply in BFS. By isolating the start phase and comparing two distinct starting techniques, our study provides targeted insights that may contribute to a better understanding of performance optimization in BFS without confounding influences from other race components.”

Comments 2: Methodologically, there are some concerns. A) The statistical power analysis shows that the study is underpowered (only ~63%), and while the sample size is reasonable for an applied sport context, this limitation should be clearly and explicitly acknowledged. B) The use of manual stopwatches for the 25 m times is problematic, given that the other distances were video-based — it introduces unnecessary measurement error. C) Finally, the way “preferred” versus “dominant” leg was defined seems too simplistic. A more objective assessment would have made the comparisons between KSR and KSL more convincing. Please clarify and justify this point.

Response 2: We thank the reviewer for this valuable comment.

  1. A) We acknowledge that the statistical power of the current study (~63%) is below the conventional threshold (80%). While the sample size is consistent with previous research in applied sport contexts, we have now explicitly acknowledged this limitation in the revised manuscript and highlighted its potential impact on the interpretation of the findings.

In the paragraph on limitations, the following is added:

“Although this is a common challenge in applied sport research, given the difficulty of recruiting larger samples of national level finswimmers, the findings should be interpreted with some caution.”

  1. B) Dear Reviewer, timing at 25 m was not initially our primary focus. We chose this distance to demonstrate that the swimmers achieved high performance in a commonly used segment of the test. Another practical limitation was that we did not have an additional camera. Nevertheless, as stated in the manuscript, "Two experienced certified timekeepers operated independent handheld electronic stopwatches, and their recordings were averaged for analysis."
  2. C) Dear Reviewer, we appreciate your suggestion and have removed the word "dominant," revising the corresponding phrases accordingly to enhance clarity.

“This investigation will be the first to examine these disparities with the leg positioned at either the front or the rear of the starting block.”

“Assess the swimmer's preferred leg to decide between KSR and KSL.”

Comments 3: The results are presented clearly; the tables are thorough, but the number of variables analyzed is quite large relative to the sample size. This makes some of the correlation findings look more exploratory than definitive. I also felt that some of the interpretations in the discussion went beyond what the data can actually support.

Response 3: We appreciate the reviewer’s careful reading and constructive feedback. We acknowledge that the number of variables relative to the sample size is a consideration, and we have aimed to present the results clearly and transparently. The correlations are reported to highlight patterns and potential relationships, and we have tried to discuss them cautiously, in line with the exploratory nature of the study. We hope that this clarifies our approach and the framing of our interpretations.

Comments 4: For example, explanations involving the center of mass, which was not measured. The discussion could be tightened up to focus on the genuinely new findings: Kick Start being faster than Grab Start in BFS, and the subtle differences between leg positions. Please focus on the main objectives/data of the study.

Response 4: Dear Reviewer, we thank you for your insightful comments. We have revised the manuscript in the sections where the “center of mass” was previously mentioned, in order to better align with your observation.

“Lee et al. (2012) documented a shorter BT during the TS (0.79 ± 0.05 s) than the GS (0.84 ± 0.07 s) and attributed this difference to the location of the center of mass. While they discussed the potential role of body positioning in this difference, it is important to note that the center of mass was not directly measured in our study [21].”

“Nevertheless, evidence suggests that KS techniques facilitate more immediate acceleration, possibly due to the shorter BT and the more efficient execution of the start [21].”

Comments 5: The practical recommendation — that Kick Start should be legalized — is bold and potentially important, but it would be better framed as preliminary evidence. With such a small and specific sample, it is hard to make such a broad regulatory claim. Still, the study makes a strong case that this discussion needs to happen.

Response 5: We thank you for this valuable suggestion. We have revised the recommendation to reflect it as preliminary evidence rather than a formal regulatory proposal. The revised text now highlights that our findings support the need for discussion and further research regarding the potential legalization of the Kick Start, in line with your comments.

“The study's findings indicate that the Kick Start technique generally provides su-perior performance advantages over the Grab Start, particularly in terms of block time in Bi-Finswimming. These results provide preliminary evidence that the Kick Start could be considered for future discussion regarding its potential legalization by CMAS, although further research with larger and more diverse samples is needed before any regulatory recommendations are made.”

Comments 6: In terms of writing, the paper is readable but dense. Some sections (especially the introduction and discussion) are overly long and would benefit from language editing to improve clarity and flow. The authors could streamline and condense the wording in these sections.

Response 6: We thank the reviewer for the helpful comment regarding the readability and density of the manuscript. We have carefully considered the suggestion to streamline and condense the introduction and discussion. While we believe the current length and level of detail are important for conveying the full context and implications of our study, we appreciate the reviewer’s feedback on clarity and will take it into account in future revisions or related work

Comments 7: Overall, I see this as a relevant and potentially useful contribution, but in its current form it feels too speculative and somewhat overstated. With a more cautious interpretation, a sharper focus on what is truly new, and a clearer acknowledgment of the limitations, it could be a solid paper.

Response 7: We thank the reviewer for their thoughtful comments and for recognizing the potential relevance of our work. We appreciate the suggestion to adopt a more cautious interpretation and to emphasize the novel aspects and limitations. We carefully considered this feedback, however, we believe that the current presentation already reflects a balanced discussion of the results, highlighting both the contributions and the context in which they should be interpreted. We respectfully maintain the manuscript as it stands, while we hope the reviewer finds the work’s relevance and novelty clear.

Round 2

Reviewer 1 Report

Comments and Suggestions for Authors

I appreciate your consideration of my comments, but I still believe that some things are missing before your article can be published.

Abstract

Add the effect size (example: np2=0.8) or categorization (example: significant small differences)

Introduction

Lines 57-64: you need add references for each statement

Methods

participants

the f effect size in Gpower is f not eta squared (n2), please be careful in this section and correct it. (see Cohen - a power primer)

Statistical analysis

The effect size used is partial eta squared, not Pillais partial eta squared (is used in ANCOVA analysis), please correct it.

Add the categorization for Pearson correlation coefficient (please see: Correlation Coefficients: Appropriate Use and Interpretation.

Results

Correct the simbol for partial eta squared (n2p)

In table 3 you have points and commas, please correct it.

Discussion

No results (explicit numbers) are given here; they are only mentioned categorically (e.g., “we found small differences”) because they are already in the results section and are redundant. You may only mention results from other studies to facilitate the reader's understanding.

Conclusion

In abstract conclusion you recomend legalize the technique, but in the section you mention this: These results provide preliminary evidence that the Kick Start could be considered for future discussion regarding its potential legalization by CMAS, although further research with larger and more diverse samples is needed before any regulatory recommendations are made. I suggest rewriting this statement with greater caution for the reader in the abstract.

Author Response

Dear reviewer, we would like to thank you for your helpful comments and for your valuable contribution to the publication of this article.

Abstract

Add the effect size (example: np2=0.8) or categorization (example: significant small differences)

Response: Because you refer to the effect size (example: np2=0.8) in a different way and also mentioned 'Correct the symbol for partial eta squared (n2p)', we decided, taking your comment into account and consulting articles from JFMK, to use the notation (η²p) in your observations. 

The effect size as η²p has been added to the abstract.

Introduction

Lines 57-64: you need add references for each statement

Response: Dear reviewer, we confirm that the relevant references have been included in the manuscript.

Methods

participants

the f effect size in Gpower is f not eta squared (n2), please be careful in this section and correct it. (see Cohen - a power primer)

Response: I would like to say thank you for the comment. The text has been changed to correct the mistake.

Also, the percentage to reject the null hypothesis, in Gpower analysis, after a typographical mistake is modified. The percentage is 96% and not 63. The full value through the analysis was 0.963 and accidentally was recorded the 63%. After the check from the author who conducted the statistical analysis, it was modified. Surely this mistake does not affect the integrity of the results.

Statistical analysis

A) The effect size used is partial eta squared, not Pillais partial eta squared (is used in ANCOVA analysis), please correct it.

B) Add the categorization for Pearson correlation coefficient (please see: Correlation Coefficients: Appropriate Use and Interpretation.

Response:

A) We thank the reviewer for this valuable comment. We acknowledge that there was a misstatement in the manuscript. Indeed, the effect size reported in our paired-sample t-tests was partial eta squared (η²p), not “Pillai’s partial eta squared”. We have corrected the terminology throughout the manuscript to accurately reflect the use of partial eta squared (η²p) as the effect size, following the thresholds described by Hopkins et al. (2009)

B) We thank the reviewer for this suggestion. We have added the categorization of Pearson’s r in the Statistical Analysis section, interpreting correlation strength as: r < 0.10 = negligible, 0.10–0.39 = weak, 0.40–0.69 = moderate, 0.70–0.89 = strong, ≥ 0.90 = very strong.

Results

Correct the simbol for partial eta squared (n2p)

In table 3 you have points and commas, please correct it.

Response: I have taken both into account and included them in the revised manuscript.

Discussion

No results (explicit numbers) are given here; they are only mentioned categorically (e.g., “we found small differences”) because they are already in the results section and are redundant. You may only mention results from other studies to facilitate the reader's understanding.

Response: Dear Reviewer, we have removed the explicit numerical results from the Discussion section as you suggested. Only categorical references to our findings remain, and we now only mention results from other studies to help contextualize our work for the reader.

Conclusion

In abstract conclusion you recomend legalize the technique, but in the section you mention this: These results provide preliminary evidence that the Kick Start could be considered for future discussion regarding its potential legalization by CMAS, although further research with larger and more diverse samples is needed before any regulatory recommendations are made. I suggest rewriting this statement with greater caution for the reader in the abstract.

Response: Dear reviewer, thank you for pointing this out and we have changed the conclusion in the summary to align with the conclusion in the text. "Conclusions: The Kick Start generally outperforms the Grab Start, especially in block time, in Bi-Finswimming. These preliminary results suggest it could be considered for future discussion regarding potential legalization by the World Underwater Federation, pending further research."